# Latency Compensated Visual-Inertial Odometry for Agile Autonomous Flight

**DOI:** 10.3390/s20082209

**Published:** 2020-04-14

**Authors:** Kyuman Lee, Eric N. Johnson

**Affiliations:** 1School of Aerospace Engineering, Georgia Institute of Technology, 270 Ferst Drive, Atlanta, GA 30313, USA; 2Faculty of Aerospace Engineering, The Pennsylvania State University, 229 Hammond Building, University Park, PA 16802, USA; eric.johnson@psu.edu

**Keywords:** VIO, UAV, EKF, IMU, camera vision, time delay, latency compensation, online temporal calibration, sensor fusion, navigation

## Abstract

In visual-inertial odometry (VIO), inertial measurement unit (IMU) dead reckoning acts as the dynamic model for flight vehicles while camera vision extracts information about the surrounding environment and determines features or points of interest. With these sensors, the most widely used algorithm for estimating vehicle and feature states for VIO is an extended Kalman filter (EKF). The design of the standard EKF does not inherently allow for time offsets between the timestamps of the IMU and vision data. In fact, sensor-related delays that arise in various realistic conditions are at least partially unknown parameters. A lack of compensation for unknown parameters often leads to a serious impact on the accuracy of VIO systems and systems like them. To compensate for the uncertainties of the unknown time delays, this study incorporates parameter estimation into feature initialization and state estimation. Moreover, computing cross-covariance and estimating delays in online temporal calibration correct residual, Jacobian, and covariance. Results from flight dataset testing validate the improved accuracy of VIO employing latency compensated filtering frameworks. The insights and methods proposed here are ultimately useful in any estimation problem (e.g., multi-sensor fusion scenarios) where compensation for partially unknown time delays can enhance performance.

## 1. Introduction

The most widely used algorithms for estimating the states of a dynamic system are a Kalman Filter [1,2] and its nonlinear versions (e.g., extended Kalman filter (EKF) [3,4] and unscented Kalman filter (UKF) [5]). The design of the standard Kalman filter does not inherently allow for significant sensor-related delays in computation. Figure 2 shows that the delay is the time difference between an instant when a measurement is taken by a sensor and another instant when the measurement is available in the filter. As an example of key delay sources, some complex sensors such as vision processors for navigation often require extensive computations to obtain higher-level information from raw sensor data. Furthermore, a closed-loop system including control logic may be an overall computational burden to a single processor. Delays resulting from heavy computation may distort the quality of state estimation since a current measurement is compared to past states of a system model. In other words, unless compensating delays in Kalman filtering, large estimation errors may accumulate over time, or even cause the filter to diverge.

The delay value is typically at least partially unknown and at least partially variable in many real applications. As an example of delay uncertainty contributors, even though a local clock is initially forced to synchronize with the centralized clock, deviations between clocks would occur because of clock drift, skew, or bias. In sensor fusion systems, when the timestamps of each sensor are typically recorded by triggered signals, non-deterministic, or non-quantized transmission delays lead to unknown time offsets on sensor streams. Moreover, if low-cost sensors such as rolling shutter cameras or software triggered devices are mounted on a vehicle, the variance of the uncertainty of timestamps might be larger. In particular, in visual-inertial odometry (VIO), we do not know the exact time instant when a camera opens and captures images for any particular pixel location. Often, exposure time depends on surrounding illumination conditions. The timestamp of the latest image by some cameras corresponds to some event such as when the shutter was triggered to start or when the entire image was available in memory. In practice, these uncertainties may be small compared to traditional sensors used for feedback in aerospace applications—but can be a major contributor to errors in emerging estimation problems such as VIO. Indeed, when estimating faster motions such as a highly agile unmanned aerial vehicle (UAV) or using progressive scan cameras, the unknown time delays may be a major driver of navigation quality and achievable controller bandwidth. We have experimentally observed the necessity of the time delay compensation to be more accurate than typically demanded, specifically for a UAV flying closed-loop on a vision-based navigation solution. With poor time delay compensation, we have observed oscillations and even divergence of estimates and closed-loop tracking error as expected, but even when we use fixed/known time delay compensation, we find time delay is still a limiting factor in accuracy and achievable control bandwidth. To illustrate, consider a UAV with a body-fixed camera that maneuvers with a more rapid rotation than reference design. Any time error produces a larger potential position estimation error with this faster rotation. Thus, even after the very best job possible has eliminated as much deterministic time delay error as is practical, we find that adapting to the non-deterministic error can enhance performance. In particular, we find it can be beneficial to deal with unknown time delays in VIO systems used in closed flight control and other systems like these.

### 1.1. Related Work

#### 1.1.1. Visual-Inertial Odometry

In recent years, an increasing demand for the research of UAVs has prompted substantial interest in VIO systems [6,7,8,9]. Delmerico and Scaramuzza [10] provide a benchmark comparison of monocular VIO algorithms for flying robots. Similar to their comparison, Table 1 illustrates state-of-the-art VIO techniques even including stereo VIO. Let us explain some relevant terms for clarity. The tightly-coupled VIO jointly optimizes over all sensor measurements (i.e., visual and inertial cost terms in VIO) within a single process which results in higher accuracy. The opposite is referred to as the loosely-coupled. Indeed, the loosely-coupled VIO does not handle the correlation of visual and inertial motion constraints, resulting in the loss of information. Moreover, at the back-end of VIO, the optimization-based VIO solves a nonlinear least-squares problem (e.g., pose-graph optimization or bundle adjustment [11]) to update a window of states, which allows for reducing errors by re-linearization [12] but with a high computational cost and possibly stuck in the local minima. In contrast, the filtering-based VIO updates only the most recent state by the Kalman filter or EKF framework, resulting in computationally faster and more efficient, but one-time linearization possibly leads to linearization errors into the estimator. For more details of the terminology, see reference [13,14].

VINS-MONO [16,21] is optimization-based visual simultaneous localization and mapping (SLAM) including loop closure. Some processes in this approach are not efficient due to the following reasons. VINS-MONO duplicates integration with the same IMU data at different timestamps for prediction and optimization purposes. That is, for publishing odometry at IMU rate, it integrates whenever IMU data arrives, whereas IMU data are also accumulated in a buffer for batch processing of integration at the time of image measurement update steps. Mourikis first introduced a multi-state constraint Kalman filter (MSCKF) [22,23], and Sun et al. [19] recently provided its stereo version. Although the real-time high-frequency VIO outputs might be crucial for UAV attitude control, MSCKF does not publish the odometry at the IMU rate but at the image rate. Furthermore, batch processing for IMU data integration in MSCKF may add redundant time delays to the filter when vision measurements are available. VINS-MONO and MSCKF are applicable to IMU and vision fusion. If we fuse other sensors such as the global positioning system (GPS) and altimeters in navigation systems, those approaches may not be operable since measurements from other sensors are available to update between images. Another limitation is that assumptions for IMU pre-integration between keyframes and backward propagation with loop closure in their approaches do not always hold. Hence, the EKF-based VIO frameworks cover a greater scope of sensor fusion problems.

Faessler et al. [17] combined semi-direct visual odometry (SVO) [24,25] with modular multi-sensor fusion (MSF) [26]. Even though this approach uses IMU data for fusing, since it is loosely-coupled, its results are sub-optimal. Paul et al. [18,27] recently proposed alternating stereo VINS that requires computation comparable to monocular VIO, yet provides scale information from the visual observations. However, this method may not be sufficient for tracking fast motion in low-latency demanding applications. Since the implementation is not open-source, this is not used for comparison in this paper. Leutenegger et al. [20] introduced a consistent keyframe-based stereo SLAM algorithm that performs nonlinear optimization over both visual and inertial cost terms. To maintain the sparsity of the system, their approach employs an approximation rendering it sub-optimal. Since it requires considerable computation resources or specific levels of sensors such as industrial grade IMUs, operating OKVIS in real-time is more challenging. Among the six algorithms in Table 1, only S-MSCKF and SVO+MSF handle an unknown time delay, so we will use their estimation results for comparison in this study.

#### 1.1.2. State Estimation Using Time-Delayed Measurements

In a number of applications, a vital problem for combining data from various sensors is the fusion of delayed observations, and if the computational delay is crucial, fusing the data in a Kalman filter is challenging. During the last 20 years, the sensor time-delay problem has been addressed by a number of methods, most of which modify the Kalman filter so that it handles delay in the sensor update step. Alexander [28] derived a method of calculating a correction term and then added it to filter estimates when lagged measurements arrive. However, because the uncertainty of measurements is often an unknown quantity until the data are processed, applying the method in time-varying systems is not addressed. To overcome the shortcoming, Larsen et al. [29] extrapolated a measurement to a current time using the past and present estimates of the Kalman filter and calculated an optimal gain for this extrapolated measurement. However, Larsen’s approach is exact for linear systems only, but if the system dynamics and measurement equations are significantly nonlinear, it can be highly inaccurate. For optimally fusing lagged sensor data in a general nonlinear system, Van Der Merwe et al. [30,31] introduced a new technique called “sample-state augmentation,” based on the Schmidt–Kalman filter [32] or stochastic cloning [33]. Appendix C provides detailed background information about the new technique. Lastly, Gopalakrishnan et al. [34] provided a survey of all previously noted methods.

All of the above methods assume that the amount of time delay is known. As an illustration, those methods only work with synchronized sensors. However, the hardware synchronization of most low-cost or customized sensors is not always available. Moreover, situations in which a current, accurate time delay might not be known can arise in real applications. To deal with the unknown time delays, Julier and Uhlmann [35] introduced the covariance union algorithm, and Sinopoli et al. [36] modeled the arrival of intermittent observations as a random variable with a probability. In addition, Choi et al. [37] and Yoon et al. [38] augmented a state vector with as many past states as the maximum number of delayed steps. The size of this augmented state vector is extremely large, and calculations with the large-size vector might require additional extensive computational effort. Recently, for the uncertainty of time delays in state estimation, Lee and Johnson [39] also suggested an approach combined with multiple-model adaptive estimation. However because of imperfect information on a certain range of the delay value, this method might not be suitable if uncertainty of time delay is high.

Instead, we directly estimate the time delay as an additional state since augmentation is a straightforward means of handling the unknown delay. Nilsson et al. [40] investigated this idea using the Taylor series expansion for small delays. However, delay values are typically larger than a time step, and the linearization in their approach does not hold for large delays. Li and Mourikis [41] also examined the state augmentation for estimating an unknown time offset between the timestamps of two sensors. However, their approach is not optimal since it performs the measurement update of delayed sensor data without the covariance correction that uses the cross-covariance term computed during the delay period. Furthermore, in the recent optimization-based method proposed by Qin and Shen [42], if cameras move at non-constant speed during the short time period like progressive scan cameras, then their assumption does not hold. Despite the short time period, the camera coordinate frame is still changing and moving. Their assumption of a constant time offset is also not general since the unknown delay may be varying. To overcome all previously noted limitations, this paper proposes a novel approach, “latency compensated filtering” based on the combined parameter-state estimator [43,44].

### 1.2. Summary of Contributions

To fuse visual measurements with unknown time delays in VIO systems and systems like them (e.g., multi-sensor fusion), the approach in this paper incorporates three correction techniques into state estimation. First, we directly estimate the unknown part of actual delays in online fashion by augmenting vehicle-feature states. With the estimated unknown part and the approximately known part of the delays, we find the most precise measurement times based on the definition of total delays introduced in this paper. Next, at the calibrated measurement time, we evaluate the Jacobian and the residual for the EKF using interpolated states. At the measurement update of the EKF, the third correction is to formulate a modified Kalman gain by the cross-covariance term computed during the delay period. The testing results of this study on flight datasets show that the proposed latency compensated VIO is a more reliable and accurate navigation solution than the existing VIO systems.

### 1.3. A Guide to This Document

The remainder of this document contains the following sections. Section 2 introduces background for all of this study. To estimate the unknown time delays and states of VIO, Section 3 and Section 4 present theory and implementation for a novel combination of the parameter estimation technique with the modified EKF that compensates delayed measurements, respectively. Section 5 shows the testing results of this study on the benchmark flight dataset. The last section concludes and plans future work.

## 2. Preliminaries

### 2.1. Sequential Measurement Update

When multiple measurements are observed at one discrete-time, sequential Kalman filtering, shown in Figure 1, is useful [45]. In fact, we obtain *N* measurements, $y_{1},y_{2},\cdots ,y_{N}$, at time *k*; that is, we first measure $y_{1}$, then $y_{2}$, ⋯, and finally $y_{N}$.

We first initialize a posteriori estimate and covariance after zero measurement is processed; that is, they are equal to the a priori estimate and covariance. For i=1,⋯,N, perform the general measurement update using the *i*-th measurement. We lastly assign the a posteriori estimate and covariance as ${\hat{x}}_{k}^{+}\leftarrow {\left({\hat{x}}_{k}\right)}_{N}$ and $P_{k}^{+}\leftarrow {\left(P_{k}\right)}_{N}$. To clarify, hat “ˆ” denotes an estimate, and superscript − and + a priori and a posteriori estimates, respectively. Based on Simon [45]’s proof that the sequential Kalman filtering is an equivalent formulation of the standard EKF, the order of updates does not affect the overall performance of estimation.

### 2.2. Vehicle Model

The nonlinear dynamics of a vehicle is driven by IMU sensor data including specific force and angular velocity. The estimated vehicle state is given by
(1)$${\hat{x}}_{V}={\left[\begin{array}{ccccc} {}^{i}{\hat{p}}_{b/i}^{T} & {}^{i}{\hat{v}}_{b/i}^{T} & {\delta \hat{\theta }}^{T} & {{\hat{b}}_{a}}^{T} & {{\hat{b}}_{\omega }}^{T} \end{array}\right]}^{T},$$
where $p_{b/i}$, $v_{b/i}$ are the position and velocity of the vehicle with respect to the inertial frame, respectively. δθ is the error quaternion of the attitude of the vehicle, and its more details are explained in references [46,47,48]. $b_{a}$, $b_{\omega }$ are the acceleration and gyroscope biases of the IMU, respectively. Left superscript *i* denotes a vector expressed in the inertial frame. The EKF propagates the vehicle state vector by dead reckoning with data from the IMU. Raw IMU sensor measurements $a_{\mathrm{raw}}$ and $\omega_{\mathrm{raw}}$ are corrupted by noise and bias as follows: (2)$$\begin{array}{cc} a_{\mathrm{raw}} & =a_{\mathrm{true}}-T_{b/i}\;{}^{i}g+b_{a}+\eta_{a},\;{\dot{b}}_{a}=\eta_{b_{a}} \end{array}$$(3)$$\begin{array}{cc} \omega_{\mathrm{raw}} & =\omega_{\mathrm{true}}+b_{\omega }+\eta_{\omega },\;{\dot{b}}_{\omega }=\eta_{b_{\omega }}, \end{array}$$
where $a_{\mathrm{true}}$, $\omega_{\mathrm{true}}$ are the true acceleration and angular rate, respectively, and *g* is the gravitational acceleration in the inertial frame. $\eta_{a}$, $\eta_{\omega }$ are zero-mean, white, Gaussian noise of the accelerometer and gyroscope measurement, and $\eta_{b_{a}}$, $\eta_{b_{\omega }}$ are the random walk rate of the acceleration and gyroscope biases. $T_{b/i}={T_{i/b}}^{T}$ denotes the rotation matrix from the inertial frame to the body frame.

The vehicle dynamics is given by
(4)$$\begin{array}{cc} {}^{i}{\dot{\hat{p}}}_{b/i} & ={}^{i}{\hat{v}}_{b/i} \end{array}$$
(5)$$\begin{array}{cc} {}^{i}{\dot{\hat{v}}}_{b/i} & ={\hat{T}}_{i/b}\;\left(\;a_{\mathrm{raw}}-{\hat{b}}_{a}\;\right)+{}^{i}g \end{array}$$
(6)$$\begin{array}{cc} {\dot{\hat{q}}}_{i/b^{\prime }} & =\frac{1}{2}\;Q\left(\;\omega_{\mathrm{raw}}-{\hat{b}}_{\omega }\;\right)\;{\hat{q}}_{i/b^{\prime }} \end{array}$$
(7)$$\begin{array}{cc} \delta \dot{\hat{\theta }} & =-\left\lceil \;(\;\omega_{\mathrm{raw}}-{\hat{b}}_{\omega }\;)\;\times \;\right\rfloor \delta \hat{\theta } \end{array}$$
(8)$$\begin{array}{cc} {\dot{\hat{b}}}_{a} & =0 \end{array}$$
(9)$$\begin{array}{cc} {\dot{\hat{b}}}_{\omega } & =0, \end{array}$$
where $\left\lceil \alpha \;\times \right\rfloor$ is a skew symmetric matrix, and function Q(·) maps a 3 by 1 vector of the angular velocity into a 4 by 4 matrix [44]. The use of the 4 by 1 quaternion representation in state estimation causes the covariance matrix to become singular, so it requires considerable accounting for the quaternion constraints. To avoid these difficulties, engineers developed the error-state Kalman filter in which 3 by 1 infinitesimal error quaternion δθ is used instead of 4 by 1 quaternion *q* in the state vector. In other words, we use attitude error quaternion $\delta q_{b/b^{\prime }}$ to express the incremental difference between tracked reference body frame $b^{\prime }$ and actual body frame *b* for the vehicle.
(10)$$\begin{array}{cc} q_{i/b} & ={\hat{q}}_{i/b^{\prime }}⊗\delta q_{b^{\prime }/b} \end{array}$$
(11)$$\begin{array}{cc} \delta q_{b^{\prime }/b} & ={\hat{q}}_{i/b^{\prime }}^{-1}⊗q_{i/b}≃\left[\begin{array}{c} 1\\ \frac{1}{2}\delta \theta \end{array}\right], \end{array}$$
where ⊗ is quaternion product defined in reference [47]. Resulting rotation matrices with error quaternion and with respect to the nominal reference body frame are
(12)$$\begin{array}{cc} T(q_{i/b}) & ={\hat{T}}_{b/i}={\hat{T}}_{b/b^{\prime }}\;{\hat{T}}_{b^{\prime }/i} \end{array}$$
(13)$$\begin{array}{cc} {\hat{T}}_{i/b^{\prime }} & ={\hat{T}}_{b^{\prime }/i}^{T}=T{\left({\hat{q}}_{i/b^{\prime }}\right)}^{T} \end{array}$$
(14)$$\begin{array}{cc} {\hat{T}}_{b^{\prime }/b} & ={\hat{T}}_{b/b^{\prime }}^{T}≃{\left(\;I+[\hat{\delta \theta }\times ]\;\right)}^{T}. \end{array}$$
Jacobian matrix $A=\frac{\partial \;\dot{x}}{\partial \;x}{|}_{\hat{x}}$ and $B=\frac{\partial \;\dot{x}}{\partial \;\eta },\;\mathrm{where}\;\eta ={\left[\;\eta_{a}^{T},\eta_{\omega }^{T},\eta_{b_{a}}^{T},\eta_{b_{\omega }}^{T}\;\right]}^{T},$ are computed in Appendix A.

### 2.3. Camera Model

An intrinsically calibrated pinhole camera model [49,50] is given by
(15)$$\begin{array}{cc} \left[\begin{array}{c} u_{j}\\ v_{j} \end{array}\right]=y_{j} & =h_{j}\left(x\right)+\zeta_{j}=\left[\begin{array}{c} f_{u}\frac{{}^{c}X_{j}}{{}^{c}Z_{j}}\;+\zeta_{u_{j}}\\ f_{v}\frac{{}^{c}Y_{j}}{{}^{c}Z_{j}}\;+\zeta_{v_{j}} \end{array}\right] \end{array}$$
(16)$$\begin{array}{cc} {\left[\begin{array}{ccc} {}^{c}X_{j}, & {}^{c}Y_{j}, & {}^{c}Z_{j} \end{array}\right]}^{T}={}^{c}p_{f_{j}/c} & =T_{c/i}\;\left(\;{}^{i}p_{f_{j}/i}-{}^{i}p_{c/i}\;\right)\\ & =T_{c/b}\;T\left(q_{i/b}\right)\;\left(\;{}^{i}p_{f_{j}/i}-{}^{i}p_{b/i}\;\right)-T_{c/b}\;{}^{b}p_{c/b} \end{array}$$
where *x* is the state vector including the vehicle and feature state, and measurement $y_{j}$ is the *j*-th feature 2*D* location on the image plane. $f_{u}$, $f_{v}$ are the horizontal and vertical focal lengths, respectively, and $\zeta_{u}$, $\zeta_{v}$ are additive, zero-mean, white, Gaussian noise of the measurement. Vectors $p_{f_{j}/c}$, $p_{f_{j}/i}$ are the *j*-th feature 3*D* position with respect to the camera frame and the inertial frame, respectively. Extrinsic parameter $T_{c/b}$ and ${}^{b}p_{c/b}$ are known and constant, and rotation matrix ${\hat{T}}_{c/i}=T_{c/b}\;{\hat{T}}_{b/b^{\prime }}\;{\hat{T}}_{b^{\prime }/i}$. Jacobian matrix $C_{j}=\frac{\partial \;y_{j}}{\partial \;x}{|}_{\hat{x}}$ is computed in Appendix A.

### 2.4. Feature Initialization

From Equation (Equation 16), if *j*-th measurement $y_{j}$ on an image is a new feature, then ${}^{i}p_{f_{j}/i}$ is unknown so it needs to be initialized. In the first step of the measurement update, we employ Gauss–Newton least-squares minimization [22,51] to estimate feature 3*D* position ${}^{i}{\hat{p}}_{f_{j}/i}$. To avoid local minima, we apply the inverse depth parameterization of the feature position [52] that is numerically more stable than the Cartesian parameterization. In other words, by the derivation explained in Appendix B, we obtain *j*-th feature 3*D* position ${}^{c_{1}}{\hat{p}}_{f_{j}/c_{1}}$ with respect to $c_{1}$ left camera frame of a stereo camera.

The *j*-th feature 3*D* position with respect to the inertial frame is
(17)$$\begin{array}{cc} {}^{i}{\hat{p}}_{f_{j}/i} & ={\hat{T}}_{i/c_{1}}\;{}^{c_{1}}{\hat{p}}_{f_{j}/c_{1}}+{}^{i}{\hat{p}}_{c_{1}/i}\\ \; & ={\hat{T}}_{i/b}\;T_{b/c_{1}}\;{}^{c_{1}}{\hat{p}}_{f_{j}/c_{1}}+\left({}^{i}{\hat{p}}_{b/i}+{\hat{T}}_{i/b}\;{}^{b}p_{c_{1}/b}\right)\\ \; & ={\hat{T}}_{i/b^{\prime }}{\hat{T}}_{b^{\prime }/b}\;\left(T_{b/c_{1}}\;{}^{c_{1}}{\hat{p}}_{f_{j}/c_{1}}+{}^{b}p_{c_{1}/b}\right)+{}^{i}{\hat{p}}_{b/i}. \end{array}$$
The new feature is initialized using only one image in which the feature is first observed. Although the new feature is initialized, since it still entails uncertainty, the EKF recursively estimates and updates its 3*D* position by augmenting into the state vector:(18)$$\begin{array}{cc} \hat{x} & ={\left[\begin{array}{cc} {\hat{x}}_{V}^{T} & {}^{i}{\hat{p}}_{f_{j}/i}^{T} \end{array}\right]}^{T}, \end{array}$$
where ${\hat{x}}_{V}$ is the estimated vehicle state vector defined in Equation (Equation 1). The overall initialization includes the initial value of the feature state and its error covariance assignment. The error covariance of the new feature are initialized using state augmentation with Jacobian *J*:(19)$$\begin{array}{cc} \left[\begin{array}{cc} P & *\\ {}* & * \end{array}\right] & =\left[\begin{array}{c} I\\ J \end{array}\right]P\left[\begin{array}{cc} I & J^{T} \end{array}\right]=\left[\begin{array}{cc} P & P\;J^{T}\\ J\;P & JPJ^{T}+P_{f_{\mathrm{new}}} \end{array}\right], \end{array}$$
where Jacobian $J=\frac{\partial \;p_{f/i}}{\partial \;x}{|}_{\hat{x}}$ is computed as follows:
(20)$$J=\left[\begin{array}{ccccc} I_{3\times 3} & 0_{3\times 3}\; & -{\hat{T}}_{i/b^{\prime }}\;\left\lceil \;(\;T_{b/c_{1}}{}^{c_{1}}{\hat{p}}_{f_{j}/c_{1}}+{}^{b}p_{c_{1}/b}\;)\;\times \;\right\rfloor \; & 0_{3\times 6} & 0_{3\times 3N_{f}} \end{array}\right].$$
$N_{f}$ is the number of all features and $P_{f_{\mathrm{new}}}$ is the initial uncertainty of the initialized new feature. The error pertains to measurement noise and the error of the least-squares minimization. In fact, since Montiel et al. [52] validate the initial uncertainty as a Gaussian distribution, the EKF including the feature initialization still holds optimality. Equations (Equation 18)–(Equation 20) arise based on “consider covariance analysis [53,54]”.

Once initialized, the EKF processes the feature state in the prediction-update loop. In the time update of the EKF, we propagate *P* by
(21)$$\begin{array}{cc} \left[\begin{array}{cc} \Phi & 0\\ 0 & I \end{array}\right]\left[\begin{array}{cc} P_{VV} & P_{Vf}\\ P_{fV} & P_{ff} \end{array}\right]\left[\begin{array}{cc} \Phi^{T} & 0\\ 0 & I \end{array}\right]+\left[\begin{array}{cc} Q_{V} & 0\\ 0 & Q_{f} \end{array}\right] & =\left[\begin{array}{cc} \Phi \;P_{VV}\;\Phi^{T}+Q_{V} & \Phi \;P_{Vf}\\ P_{fV}\;\Phi^{T} & P_{ff}+Q_{f} \end{array}\right], \end{array}$$
where state transition matrix Φ≈I+AΔt. $P_{VV}:=E\left[\;\left(x_{V}-{\hat{x}}_{V}\right){\left(x_{V}-{\hat{x}}_{V}\right)}^{T}\;\right]$, $P_{ff}$ is the error covariance of all features, and $P_{Vf}=P_{fV}^{T}$ represents vehicle–feature correlations. In addition, we assume that the surroundings are static, so the dynamics of features ${\dot{\hat{p}}}_{f_{j}/i}=0$. In the measurement update of the EKF, only tracked features are used for the update. For the efficient management of the map database, if the size of the state vector exceeds than the maximum limit, then the feature with the least number of observations is pruned and marginalized.

## 3. Theory

### 3.1. Definition of Time Delays

Based on dead reckoning, the EKF propagates state *x* and its error covariance *P* at time *t* when IMU sensor data $a_{\mathrm{raw}}$ and $\omega_{\mathrm{raw}}$ are measured. Since an IMU is a discrete-time sensor, the time update of the EKF is processed in discrete time step $k=\left(\mathrm{integer}\right)\;\left(t\;/\;\Delta \;t_{\mathrm{IMU}}\right),$ where (integer) defines the conversion of all data types to integers, continuous time $t\in [0,t_{\mathrm{final}}]$, and $\Delta \;t_{\mathrm{IMU}}$ is the sampling rate of the IMU. $\Delta \;t_{\mathrm{IMU}}$ is generally almost constant since a micro controller such as Arduino and Pixhawk calculates precise timestamps in milliseconds for each IMU measurement. Next, whenever new vision data from an image are arrived at the filter, the EKF performs the measurement update for correcting the state estimate and its error covariance. As introduced in Section 1, various reasons such as image processing produce time delays that the time stamps of vision data contain. For clarity, this section defines the time delay in detail.

Latency is the time difference between when an image was grabbed and when vision data from the image are updated in the filter, shown in Figure 2.

That is, true delays $\Delta \;t_{d}$ are written as
(22)$$t=t_{\mathrm{img}}+\Delta \;t_{d},$$
where *t* is current IMU time and $t_{\mathrm{img}}$ is the time when the current image was captured. In essence, we treat IMU time as our common time reference, and we do not necessarily know the exact time when images are grabbed. The timestamp of each image is encoded by indirect ways such as triggers. In other words, true image time $t_{\mathrm{img}}$ constitutes readable timestamps $t_{\mathrm{img},\;\mathrm{raw}}$ and unknown $\delta t_{d}$. Let us define time differences $\Delta \;{\bar{t}}_{d}$ between the time readouts of sensors as follows: (23)$$\begin{array}{cc} t_{\mathrm{img},\;\mathrm{raw}} & =t_{\mathrm{img}}+\delta t_{d} \end{array}$$(24)$$\begin{array}{cc} \Delta \;{\bar{t}}_{d} & :=t-t_{\mathrm{img},\;\mathrm{raw}}=\Delta \;t_{d}-\delta t_{d} \end{array}$$(25)$$\begin{array}{cc} \Delta \;t_{d} & =\Delta \;{\bar{t}}_{d}+\delta t_{d}, \end{array}$$
where $\Delta \;{\bar{t}}_{d}$ and $\delta t_{d}$ are the approximately known and the unknown parts of true delays $t_{d}$, respectively.

### 3.2. Approximately Known Part of Time Delays

$\Delta \;{\bar{t}}_{d}$ is either a fixed value determined by offline beforehand tuning or readable differences between the time stamps of image and the time stamps of IMU data. Indeed, regardless of a constant value or readable varying delays, approximate delay $\Delta \;{\bar{t}}_{d}$ is a known value. Let the discrete steps of the approximately known part be $d=\left(\mathrm{integer}\right)\;\left(\Delta \;{\bar{t}}_{d}\;/\;\Delta \;t_{\mathrm{IMU}}\right)$, where (integer) means type conversion to integer from other types; that is, *d* is the quotient of division $\frac{\Delta \;{\bar{t}}_{d}}{\Delta \;t_{\mathrm{IMU}}}$.

#### 3.2.1. Jacobian and Residual—“Baseline Correction”

Since $\delta t_{d}$ is unknown, we first consider only the $\Delta \;{\bar{t}}_{d}$ term as our delay of the system. From the system models given in Section 2.2 and Section 2.3, only measurements from the camera model depend on the time delays. To correct the Jacobian and residual with approximately known delays, interpolation and quaternion slerp are required. Since $k-d\neq \left(\mathrm{integer}\right)\left(\frac{t-\Delta \;{\bar{t}}_{d}}{\Delta t_{\mathrm{IMU}}}\right)$, we define new time notation $\left[\;k-\bar{d}\;\right]$ as
$$\left[\;k-\bar{d}\;\right]:=\frac{t_{\mathrm{img},\;\mathrm{raw}}}{\Delta t_{\mathrm{IMU}}}=\frac{t-\Delta \;{\bar{t}}_{d}}{\Delta t_{\mathrm{IMU}}}.$$
When time $\left[\;k-\bar{d}\;\right]$ is expressed at subscript (e.g., $x_{\left[\;k-\bar{d}\;\right]}$, $P_{\left[\;k-\bar{d}\;\right]}$), we will use the shorthand notation without [] (e.g., $x_{k-\bar{d}}$, $P_{k-\bar{d}}$).

Although delay *d* in discrete-time systems is the number of delayed samples, time $\left[\;k-\bar{d}\;\right]$ is not required to be an integer by reading timestamps of each sensor. Since $\left[\;k-\bar{d}\;\right]$ is not an integer, we cannot directly access the values of either ${\hat{x}}_{k-\bar{d}}$ or its corresponding error covariance $P_{k-\bar{d}}$, so relevant interpolation is required instead.

Mathematically, linear interpolation constructs a new data point within the range of two known adjacent data points by the same slope of two lines [55]. Let us take the nearest integer time step k−d, which is greater than or equal to $\left[\;k-\bar{d}\;\right]$, shown in Figure 3a.

With two data points, either $\left(k-d-1,\;{\hat{x}}_{k-d-1}\right)$ and $\left(k-d,\;{\hat{x}}_{k-d}\right)$ or $\left(k-d-1,\;P_{k-d-1}\right)$ and $\left(k-d,\;P_{k-d}\right)$, the interpolants at time $\left[\;k-\bar{d}\;\right]$ are given by
$$\begin{array}{cc} {\hat{x}}_{k-d}-{\hat{x}}_{k-d-1} & =\frac{{\hat{x}}_{k-d}-{\hat{x}}_{k-\bar{d}}}{k-d-[\;k-\bar{d}\;]} \end{array}$$
(26)$$\begin{array}{cc} {\hat{x}}_{k-d}-{\hat{x}}_{k-\bar{d}} & ≃\left(\frac{t}{\Delta \;t_{\mathrm{IMU}}}-d-\frac{t-\Delta \;{\bar{t}}_{d}}{\Delta t_{\mathrm{IMU}}}\right)\left({\hat{x}}_{k-d}-{\hat{x}}_{k-d-1}\right) \end{array}$$
(27)$$\begin{array}{cc} {\hat{x}}_{k-\bar{d}} & =\left(1-\frac{\Delta \;{\bar{t}}_{d}}{\Delta t_{\mathrm{IMU}}}+d\right){\hat{x}}_{k-d}+\left(\frac{\Delta \;{\bar{t}}_{d}}{\Delta t_{\mathrm{IMU}}}-d\right){\hat{x}}_{k-d-1}, \end{array}$$
where $k=\left(\mathrm{integer}\right)\left(\frac{t}{\Delta \;t_{\mathrm{IMU}}}\right)\approx \frac{t}{\Delta \;t_{\mathrm{IMU}}}$ in Equation (Equation 26). Likewise,
$$P_{k-\bar{d}}=\left(1-\frac{\Delta \;{\bar{t}}_{d}}{\Delta t_{\mathrm{IMU}}}+d\right)P_{k-d}+\left(\frac{\Delta \;{\bar{t}}_{d}}{\Delta t_{\mathrm{IMU}}}-d\right)P_{k-d-1}.$$

Although we compute the interpolants at time $\left[\;k-\bar{d}\;\right]$ using linear interpolation because of the constraint and specialty of quaternion, another interpolation is required. Slerp is shorthand for spherical linear interpolation, introduced by Ken Shoemake [56] in the context of quaternion interpolation for the purpose of animating 3*D* rotation. Interpolants refer to constant-speed motion along a unit-radius circle arc, shown in Figure 3b. Based on the fact that any point on the curve is linear combination of the given ends, the geometric formula [56,57] is
(28)$$\begin{array}{cc} \Theta & =\cos^{-1}\left(q_{k-d}\cdot q_{k-d+1}\right) \end{array}$$
(29)$$\begin{array}{cc} {\hat{q}}_{k-\bar{d}} & =\frac{\sin\left[\left(1-\frac{\Delta \;{\bar{t}}_{d}}{\Delta t_{\mathrm{IMU}}}+d\right)\Theta \right]}{\sin\Theta }\;{\hat{q}}_{k-d}+\frac{\sin\left[\left(\frac{\Delta \;{\bar{t}}_{d}}{\Delta t_{\mathrm{IMU}}}-d\right)\Theta \right]}{\sin\Theta }\;{\hat{q}}_{k-d-1}, \end{array}$$
where since only unit quaternions are valid rotations, normalization of each quaternion before applying Slerp is a prerequisite.

Θ is a smaller angle between two end quaternions, so we ensure that −90deg≤Θ≤90deg. If the dot product in Equation (Equation 28) is negative, Slerp does not represent the shortest path. To prevent long paths, we negate one of end quaternions since *q* and −q are equivalent when the negation is applied to all four components. If two quaternions input $q_{k-d}$, $q_{k-d+1}$ are too close, then interpolants by linear interpolation are acceptable. Otherwise, $\cos^{-1}\left(\cdot \right)$ in Equation (Equation 28) is safe computation because the dot product is in the range of the threshold.

With suitable interpolants at time $\left[\;k-\bar{d}\;\right]$, a baseline approach modifies the feature initialization in Appendix B and the measurement update. At time *k*, the vision data of an image grabbed at time $\left(t-\Delta \;t_{d}\right)$ arrive at the filter for either the feature initializations or the sequential measurement updates. If *j*-th measurement $y_{j}$ on the last image is a new feature, then, from Equations (Equation 18) and (Equation 19), state $\hat{x}$ and covariance *P* at current time *k* are augmented as follows: (30)$$\begin{array}{cc} {\hat{x}}_{k} & \overset{\mathrm{aug}}{\Rightarrow }{\left[\begin{array}{cc} {\hat{x}}_{k}^{T} & {}^{i}{\hat{p}}_{f_{j}/i}^{T} \end{array}\right]}^{T}\; \end{array}$$(31)$$\begin{array}{cc} P_{k} & \overset{\mathrm{aug}}{\Rightarrow }\left[\begin{array}{cc} P_{k} & P_{k}\;{\left(J_{j}\right)}_{k-\bar{d}}^{T}\\ {\left(J_{j}\right)}_{k-\bar{d}}\;P_{k} & {\left(J_{j}\right)}_{k-\bar{d}}\;P_{k}\;{\left(J_{j}\right)}_{k-\bar{d}}^{T}+P_{f_{j_{\mathrm{new}}}} \end{array}\right]\; \end{array}$$
where ${}^{i}{\hat{p}}_{f_{j}/i}=T_{i/b}{|}_{k-\bar{d}}\;\;{}^{b}{\hat{p}}_{f_{j}/b}+{\left({}^{i}p_{b/i}\right)}_{k-\bar{d}}$ and ${\left(J_{j}\right)}_{k-\bar{d}}=\frac{\partial p_{f_{j}/i}}{\partial x}{|}_{{\hat{x}}_{k-\bar{d}}}$. ${}^{b}{\hat{p}}_{f_{j}/b}$ is initialized by Gaussian-Newton least-squares minimization derived in Appendix B. Although we assume static features, since the feature initialization is related to estimated camera pose at the time when the delays begin, corrected Jacobian $J_{j}$ is required in the initialization steps.

If *j*-th measurement $y_{j}$ on the image is a tracked feature, then we correct only residual *r* and Jacobian *C* in the following measurement update: (32)$$\begin{array}{cc} K_{j} & ={\left(P_{k}\right)}_{j-1}\;{\left(C_{j}\right)}_{k-\bar{d}}^{T}\;{\left(\;{\left(C_{j}\right)}_{k-\bar{d}}\;{\left(P_{k}\right)}_{j-1}\;{\left(C_{j}\right)}_{k-\bar{d}}^{T}+R\;\right)}^{-1} \end{array}$$(33)$$\begin{array}{cc} {\left({\hat{x}}_{k}\right)}_{j} & ={\left({\hat{x}}_{k}\right)}_{j-1}+K_{j}\;\left(\;y_{j}{|}_{t-\Delta \;t_{d}}\;-h_{j}\left({\hat{x}}_{k-\bar{d}}\right)\;\right) \end{array}$$(34)$$\begin{array}{cc} {\left(P_{k}\right)}_{j} & ={\left(P_{k}\right)}_{j-1}-K_{j}\;{\left(C_{j}\right)}_{k-\bar{d}}\;{\left(P_{k}\right)}_{j-1} \end{array}$$
where corrected residual ${\left(r_{j}\right)}_{k-\bar{d}}=y_{j}{|}_{t-\Delta \;t_{d}}\;-h_{j}\left({\hat{x}}_{k-\bar{d}}\right)$ and Jacobian ${\left(C_{j}\right)}_{k-\bar{d}}=\frac{\partial \;h_{j}\left(x\right)}{\partial \;x}{|}_{{\hat{x}}_{k-\bar{d}}}$. *R* is the measurement noise covariance of $y_{j}{|}_{t-\Delta \;t_{d}}$, and $K_{j}$ is sub-optimal Kalman gain computed by current covariance. As sequential Kalman Filtering introduced in Section 2.1, if *j* is the first feature on the current image (i.e., *j* = 0), then assign ${\left({\hat{x}}_{k}\right)}_{0}\leftarrow {\hat{x}}_{k}^{-}$, ${\left(P_{k}\right)}_{0}\leftarrow P_{k}^{-}$, and if *j* is the last feature on the current image (i.e., *j* = $N_{k}$), then assign ${\hat{x}}_{k}^{+}\leftarrow {\left({\hat{x}}_{k}\right)}_{N}$, $P_{k}^{+}\leftarrow {\left(P_{k}\right)}_{N}$. Before measurement updates (Equation 32)–(34), a chi-squared gating test rejects outliers of each measurement. For only this test purpose in the case of baseline correction, we add uncertainty due to time delay. Procedures in Equations (Equation 30)–(34) are referred to as the “baseline correction.”

#### 3.2.2. Cross-Covariance—“Covariance Correction”

During the delay period, even though an image was already captured in the past, since vision data from the image have not yet arrived at the filter, the EKF is not able to perform the measurement update. Indeed, the filter processes only time update. When a vision data packet from the image finally arrives and is ready to update in the filter, we simply execute the Jacobian and residual correction in Equations (Equation 32)–(34) using the delayed measurements. However, unlike the baseline correction, if the filter updates as if the measurements arrive immediately without delays (like red lines in Figure 4), then filter can achieve a more accurate estimate. In fact, covariance correction presented in this section (like blue lines in Figure 4) is as if the filter accomplished the general measurement update at the time instant when the image was captured. In other words, red lines in Figure 4 are ideal but unrealistic, and blue lines in the figure are practical. The red lines process the measurement update first and then time update; however, the order of the processes of the blue lines is the opposite. Only the order of the processes has changed.

Among a variety of fusing techniques for time-delayed observations discussed in Section 1.1.2, the stochastic cloning [33]-based method (i.e., the Schmidt EKF [30,31]) is applicable to varying delays and nonlinear functions such as the vehicle and camera models described in Section 2.2 and Section 2.3, respectively. Thus, this study modifies the method for finding the optimal navigation solution of vision-aided inertial navigation systems.

Let us introduce new notation $P^{\mathrm{dly}}$. $P^{\mathrm{dly}}$ is *P* covariance matrix at the time when the true delays begin. In the scope of this section, $P^{\mathrm{dly}}≃P_{k-\bar{d}}$. In addition, when this section uses corrected residual ${\left(r_{j}\right)}_{k-\bar{d}}$ and Jacobians ${\left(J_{j}\right)}_{k-\bar{d}}$, ${\left(C_{j}\right)}_{k-\bar{d}}$, we will use their shorthand notations as $r_{j}$ and $J_{j}$, $C_{j}$, respectively. That is, each residual and Jacobian is corrected based on Section 3.2.1. In addition to the baseline correction, we correct error covariance in both the feature initialization and the measurement update when delayed vision data are available in the filter.

If *j*-th feature measurement $y_{j}$ on the recent image is a new feature, the augmentation of $P^{\mathrm{dly}}$ in the feature initialization is similar to Equation (31). On the other hand, since Jacobian $J_{j}$ is computed at the time when the delays begin, the augmentation of covariance matrix $P_{k}$ at the current time is as follows in a different way: (35)$$\begin{array}{cc} P_{k} & \overset{\mathrm{aug}}{\Rightarrow }\left[\begin{array}{cc} P_{k} & 0\\ 0 & d\;Q_{f} \end{array}\right] \end{array}$$(36)$$\begin{array}{cc} P^{\mathrm{dly}} & \overset{\mathrm{aug}}{\Rightarrow }\left[\begin{array}{cc} P^{\mathrm{dly}} & P^{\mathrm{dly}}\;J_{j}^{T}\\ J_{j}\;P^{\mathrm{dly}} & J_{j}\;P^{\mathrm{dly}}\;J_{j}^{T}+P_{f_{j_{\mathrm{new}}}} \end{array}\right], \end{array}$$
where $d=\left(\mathrm{integer}\right)\;\frac{\Delta \;{\bar{t}}_{d}}{\Delta \;t_{\mathrm{IMU}}}$ and $J_{j}=\frac{\partial \;p_{f_{j}/i}}{\partial x}{|}_{{\hat{x}}_{k-\bar{d}}}$. State estimate ${\hat{x}}_{k}$ is augmented by Equation (Equation 30).

When *j*-th delayed vision data $y_{j}$ is ready to update at time *k*, we modify the measurement update steps of the sequential Kalman filtering as follows: $$\begin{array}{cc} S_{j} & =C_{j}\;{\left(P^{\mathrm{dly}}\right)}_{j-1}\;C_{j}^{T}+R \end{array}$$(37)$$\begin{array}{cc} K_{j}^{\mathrm{crs}} & ={\left(P^{\mathrm{crs}}\right)}_{j-1}\;C_{j}^{T}\;S_{j}^{-1} \end{array}$$(38)$$\begin{array}{cc} {\left({\hat{x}}_{k}\right)}_{j} & ={\left({\hat{x}}_{k}\right)}_{j-1}+K_{j}^{\mathrm{crs}}\;r_{j} \end{array}$$(39)$$\begin{array}{cc} {\left(P_{k}\right)}_{j} & ={\left(P_{k}\right)}_{j-1}-K_{j}^{\mathrm{crs}}\;C_{j}\;{\left(P^{\mathrm{crs}}\right)}_{j-1}^{T}, \end{array}$$
where $r_{j}=y_{j}{|}_{t-\Delta \;t_{d}}\;-h_{j}\left({\hat{x}}_{k-\bar{d}}\right)$ and $C_{j}=\frac{\partial \;h_{j}\left(x\right)}{\partial \;x}{|}_{{\hat{x}}_{k-\bar{d}}}$. $P^{\mathrm{crs}}$ is the relevant cross-covariance term during the delay period. This term, which fuses a current prediction of the state with an observation related to the lagged state of the system, is used for formulating modified Kalman gain matrix $K^{\mathrm{crs}}$. Equation (39) still holds Joseph’s form [58] that preserves the symmetry of the updated covariance and ensures its the positive definiteness. By sequential update provisions, the state estimate and covariance at time $\left[\;k-\bar{d}\;\right]$ are also updated as follows: (40)$$\begin{array}{cc} K_{j}^{\mathrm{dly}} & ={\left(P^{\mathrm{dly}}\right)}_{j-1}\;C_{j}^{T}\;S_{j}^{-1} \end{array}$$(41)$$\begin{array}{cc} {\left({\hat{x}}_{k-\bar{d}}\right)}_{j} & ={\left({\hat{x}}_{k-\bar{d}}\right)}_{j-1}+K_{j}^{\mathrm{dly}}\;r_{j} \end{array}$$(42)$$\begin{array}{cc} {\left(P^{\mathrm{dly}}\right)}_{j} & ={\left(P^{\mathrm{dly}}\right)}_{j-1}-K_{j}^{\mathrm{dly}}\;C_{j}\;{\left(P^{\mathrm{dly}}\right)}_{j-1}. \end{array}$$

At time $t_{\mathrm{img}}$, when cameras open for capturing the image, the cross-covariance matrix is initialized with covariance at that time; that is, $P^{\;\mathrm{crs}}\leftarrow P^{\;\mathrm{dly}}\approx P_{k-\bar{d}}$. During the delay period, from time $\left[\;k-\bar{d}\;\right]$ to current time *k*, if no other measurements are fused into the filter, the cross-covariance is only propagated by the following computation based on the Schmidt–Kalman filter [30,32,33]:(43)$$\begin{array}{cc} \Phi^{\;\mathrm{crs}} & =\prod_{i=k-1}^{k-d}\left[\begin{array}{cc} \Phi_{i} & 0\\ 0 & I \end{array}\right]=\left[\begin{array}{cc} \left(\prod_{i=k-1}^{k-d}\Phi_{i}\right) & 0\\ 0 & I \end{array}\right] \end{array}$$(44)$$\begin{array}{cc} P^{\mathrm{crs}} & =\Phi^{\mathrm{crs}}\;P^{\mathrm{dly}} \end{array}$$(45)$$\begin{array}{cc} \; & =\left[\begin{array}{cc} \left(\prod_{i=k-1}^{k-d}\Phi_{i}\right)\;P_{vv}^{\mathrm{dly}} & \left(\prod_{i=k-1}^{k-d}\Phi_{i}\right)\;P_{vf}^{\mathrm{dly}}\\ P_{fv}^{\mathrm{dly}} & P_{ff}^{\mathrm{dly}} \end{array}\right] \end{array}$$
where Φ is the state transition matrix. In the sequential measurement update, based on updated ${\left(P^{\mathrm{dly}}\right)}_{j}$ in Equation (42), updating ${\left(P^{\mathrm{crs}}\right)}_{j-1}$ is straightforward as follows:(46)$${\left(P^{\mathrm{crs}}\right)}_{j}=\Phi^{\;\mathrm{crs}}\;{\left(P^{\mathrm{dly}}\right)}_{j}$$

If other measurements from other sensors such as an altimeter and GPS are fused during the delay period, then $P^{\mathrm{dly}}$ and cross-covariance $P^{\mathrm{crs}}$ are also recursively updated using the Kalman gain of the other measurements. For this case, Equations (Equation 43)–(Equation 46) do not hold any longer. For more details, see Appendix C. All modification in this section is referred to as “covariance correction.” Furthermore, the optimality of this covariance correction is guaranteed based on the fact that the standard Kalman filter is an optimal filter since Appendix C proves that the covariance correction is identical to the standard EKF. Hence, the proposed correction still holds its optimality. Section 4 will describe its practical implementation.

### 3.3. Unknown Part of Time Delays—“Online Calibration”

Although residual, Jacobians, and covariance are corrected for measurements with time delays, if $\Delta \;{\bar{t}}_{d}$ is uncertain readouts or $\delta t_{d}$ is the larger portion of true delays, we cannot guarantee the reliability of the correction algorithm (Figure 5). For the robustness of vision-aided navigation systems, we need to additionally investigate the unknown part of true delays.

Figure 5 shows three corrections in the latency compensated VIO presented in this study. From the standard Kalman filter, if one does not account for time delay, propagation, and measurement update look like grey lines in Figure 5. For the last correction, we estimate the unknown part of time delays to obtain more precise time instant when the delays begin. As discussed in Section 1, unknown phenomena such as clock bias, drift, skews, and asynchronization cause $\delta t_{d}$, so $\delta t_{d}$ may be a positive or negative value.

State estimation theory can be used to estimate not only the states but also the unknown parameters of the system [59]. Numerous researchers [60,61,62] have proved that state augmentation functions are easy to use with state observers, so we enable design a state observer by state augmentation to estimate the unknown part of the time delays. To estimate unknown delay value $\delta t_{d}$, we first augment state estimates ${\hat{x}}_{V}$ and covariance $P_{vv}$ of the vehicle as follows: (47)$$x_{V}\overset{\mathrm{aug}}{\Rightarrow }{\left[\begin{array}{cc} x_{V}^{T}, & \delta t_{d} \end{array}\right]}^{T},\;\;\;P_{vv}\overset{\mathrm{aug}}{\Rightarrow }\left[\begin{array}{cc} P_{vv} & P_{v\;\delta t_{d}}\\ P_{\delta t_{d}\;v} & P_{\delta t_{d}} \end{array}\right].$$
Like the modeling of the IMU biases in Equations (Equation 2) and (3), we model the dynamics of $\delta t_{d}$ using a small artificial noise term
(48)$${\dot{\delta t}}_{d}=\eta_{d},\;\;\;\;{\dot{\hat{\delta t}}}_{d}=0,$$
where $\eta_{p}$ is a random walk rate that allows the EKF to change its estimate of $\delta t_{d}$; that is, the power spectral density of $\eta_{p}$ represents the variability of ${\delta t}_{d}$. In fact, this is a conventional random walk model for an unknown parameter that may be varying—commonly seen for things like gyro bias, as done here. If additional modeling information about the way time delays is expected to vary is known, then it could be captured here with a more complex model.

Let us rewrite the definition of time delays:$$\begin{array}{cc} t-t_{\mathrm{img}} & =\Delta \;t_{d}\\ \Delta \;t_{d} & =\Delta \;{\bar{t}}_{d}+\delta t_{d}=\left(t-t_{\mathrm{img},\;\mathrm{raw}}\right)+\delta t_{d}. \end{array}$$
For clarity, we define new time notation $\left[\;k-\hat{d}\;\right]$ as
$$\begin{array}{cc} \left[\;k-\hat{d}\;\right] & :=\frac{t_{\mathrm{img},\;\mathrm{raw}}-{\hat{\delta t}}_{d}}{\Delta t_{\mathrm{IMU}}}\approx \frac{t_{\mathrm{img}}}{\Delta t_{\mathrm{IMU}}}\\ & =\frac{t-(\Delta \;{\bar{t}}_{d}+{\hat{\delta t}}_{d})}{\Delta t_{\mathrm{IMU}}}, \end{array}$$
where now time $\left[\;k-\hat{d}\;\right]$ is the most precise time instant when the image was captured. To apply the relevant interpolation techniques to the state estimates and covariance at time $\left[\;k-\hat{d}\;\right]$, we access their values at the nearest integer time step k−s, where $s=\left(\mathrm{integer}\right)\;\left(\Delta \;{\bar{t}}_{d}+{\hat{\delta t}}_{d}\right)/\Delta t_{\mathrm{IMU}}$. In other words, *s*, discrete delayed samples including estimated latency, is greater than or equal to $\left[\;k-\hat{d}\;\right]$, as shown in Figure 2.

To operate the augmented system, we match its dimension by augmenting other matrices. In the time update, since ${\dot{\hat{\delta t}}}_{d}=0$, the state transition matrix and the process noise covariance matrix are augmented
(49)$$\Phi \overset{\mathrm{aug}}{\Rightarrow }\left[\begin{array}{cc} \Phi & 0\\ 0 & I \end{array}\right],\;\;\;\;Q_{v}\overset{\mathrm{aug}}{\Rightarrow }\left[\begin{array}{cc} Q_{v} & 0\\ 0 & Q_{d} \end{array}\right]$$
where *I* is due to ${\dot{\hat{\delta t}}}_{d}=0$ and the Gaussian white noise $\eta_{d}∼N\left(0,Q_{d}\right)$. Under the assumption of static features, since estimated latency $\delta t_{d}$ pertains to only vision measurements, we compute augmented elements of Jacobian matrices *J* and *C* [41,43]. In fact, from Equation (Equation 20), Jacobian $J_{j}$ in the feature initialization is augmented as follows: $${\left(J_{j}\right)}_{k-\hat{d}}\overset{\mathrm{aug}}{\Rightarrow }\;\left[\begin{array}{cccccc} I_{3\times 3} & 0_{3\times 3}\; & \frac{\partial \;{\hat{p}}_{f_{j}/i}}{\partial \;\delta \hat{\theta }}{|}_{\left[\;k-\hat{d}\;\right]}\; & 0_{3\times 6} & J_{j_{\delta t_{d}}}\; & |0\cdots \end{array}\right],$$
where
(50)$$\begin{array}{cc} J_{j_{\delta t_{d}}} & =\frac{\partial \;{\hat{p}}_{f_{j}/i}}{\partial \;{\hat{\delta t}}_{d}}{|}_{\left[\;k-\hat{d}\;\right]}≃\frac{\partial \;{\hat{p}}_{f_{j}/i}}{\partial \;x}{|}_{{\hat{x}}_{k-\hat{d}}}\cdot \frac{\partial \;x}{\partial \;t}{|}_{\left[\;k-\hat{d}\;\right]}\cdot \frac{\partial \;t}{\partial \;{\hat{\delta t}}_{d}}{|}_{\left[\;k-\hat{d}\;\right]}\;\\ \; & ={\left(J_{j}\right)}_{k-\hat{d}}\;{\dot{\hat{x}}}_{k-\hat{d}}\;\\ \; & =\left[\begin{array}{cccccc} I_{3\times 3} & 0_{3\times 3}\; & \frac{\partial \;{\hat{p}}_{f_{j}/i}}{\partial \;\delta \hat{\theta }}{|}_{\left[\;k-\hat{d}\;\right]}\; & 0_{3\times 3} & 0_{3\times 3}\; & |0\cdots \end{array}\right]\;{\left[\begin{array}{c} {}^{i}{\dot{\hat{p}}}_{b/i}\\ {}^{i}{\dot{\hat{v}}}_{b/i}\\ \delta \dot{\hat{\theta }}\\ 0\\ 0\\ {\left(0\cdots 0\right)}^{T} \end{array}\right]}_{\left[\;k-\hat{d}\;\right]}\;\\ \; & ={\left({}^{i}{\hat{v}}_{b/i}\right)}_{k-\hat{d}}\;+\;\left(\frac{\partial \;{\hat{p}}_{f_{j}/i}}{\partial \;\delta \hat{\theta }}{|}_{\left[\;k-\hat{d}\;\right]}\right)\;{\delta \dot{\hat{\theta }}}_{k-\hat{d}} \end{array}$$
Furthermore, from Equation (Equation 53), augmented Jacobian $C_{j}$ in the measurement update is
$${\left(C_{j}\right)}_{k-\hat{d}}\overset{\mathrm{aug}}{\Rightarrow }{\left[\begin{array}{cccccccccc} \frac{\partial \;y_{j}}{\partial \;{}^{i}{\hat{p}}_{b/i}} & 0 & \frac{\partial \;y_{j}}{\partial \;\delta \hat{\theta }} & 0 & 0 & \; & \frac{\partial \;y_{j}}{\partial \;{\hat{\delta t}}_{d}}\; & |\;0\cdots & \frac{\partial \;y_{j}}{\partial \;{}^{i}{\hat{p}}_{f_{j}/i}} & \cdots 0 \end{array}\right]}_{\left[\;k-\hat{d}\;\right]},$$
where
(51)$$\begin{array}{cc} \frac{\partial \;y_{j}}{\partial \;{\hat{\delta t}}_{d}}{|}_{\left[\;k-\hat{d}\;\right]}\; & ≃\frac{\partial \;y_{j}}{\partial \;x}{|}_{{\hat{x}}_{k-\hat{d}}}\cdot \frac{\partial \;x}{\partial \;t}{|}_{\left[\;k-\hat{d}\;\right]}\cdot \frac{\partial \;t}{\partial \;{\hat{\delta t}}_{d}}{|}_{\left[\;k-\hat{d}\;\right]}={\left(C_{j}\right)}_{k-\hat{d}}\;{\dot{\hat{x}}}_{k-\hat{d}}\;\\ \; & =\left(\frac{\partial \;y_{j}}{\partial \;{}^{i}{\hat{p}}_{b/i}}{|}_{\left[\;k-\hat{d}\;\right]}\right)\;{\left({}^{i}{\hat{v}}_{b/i}\right)}_{k-\hat{d}}\;+\;\left(\frac{\partial \;y_{j}}{\partial \;\delta \hat{\theta }}{|}_{\left[\;k-\hat{d}\;\right]}\right)\;{\delta \dot{\hat{\theta }}}_{k-\hat{d}}. \end{array}$$
Here, let us call the combination of the estimation of the unknown latency in this section with the baseline correction “online calibration.” Therefore, to reliably estimate the state variable and effectively compensate the total delays, we incorporate all three corrections, called “latency-adaptive filtering.”

## 4. Implementation

This section summarizes and describes an implementation of the proposed method. Figure 6 illustrates a flow chart of the overall process.

### 4.1. Forward Computation of Cross-Covariance

Even though delays begins $\Delta t_{d}$ time prior, estimated delay value ${\hat{t}}_{d}$ is only accessible when delay finished. That is, during the delay period from $t_{\mathrm{img}}$ to *t*, $\Delta {\hat{t}}_{d}$ is unknown yet. $\Delta {\hat{t}}_{d}$ is estimated at current time *k*. Since estimated delay value $\Delta {\hat{t}}_{d}$ is unknown up to time *k*, we are not sure when the covariance correction begins computing cross-covariance $P^{\mathrm{crs}}$. Theoretically, when $\Delta {\hat{t}}_{d}$ is estimated at time *k*, we compute $P^{\mathrm{dly}}$ and $P^{\mathrm{crs}}$ by backward from time *k* to time $\left[\;k-\hat{d}\;\right]$ with saved Jacobians and covariance during the delay period. This is an ideal computation, but not realistic. Backward computation that is used in [43] is impossible for real-time operations since storing large matrices such as sequences of Jacobian and covariance matrices allocates huge memory uses. Furthermore, backward computing is not efficient because it iterates backward at time *k* like batch processing.

Instead, for a real-time framework, an approximated way of forward computation of cross-covariance is introduced. Since ${\hat{\delta t}}_{d}=0$, we assume that the time delay does not change in state propagation during the delay period, so a posteriori estimate of time delay when the last measurement update is assumed to be a priori estimate of the delay at the current time. Next, under this assumption, we predict when the time delay of the next image begins. At the predicted time instant, we store the covariance matrix once for $P^{\mathrm{dly}}$ and recursively calculate $\Phi^{\;\mathrm{crs}}$ for $P^{\mathrm{crs}}$.

### 4.2. Summarized Algorithm

When the size of the state after augmentation in the feature initialization steps exceeds a maximum threshold, we prune the number of features in the database. The system in this study finds an index for the best place to insert a new point in the database. The one with the least number of observations or frequent outliers is marginalized. Unlike Lee at al. [43], this study does not estimate the total parts of time delays, so the latency compensated filter does not entail specific constraints. That is, this study estimates only unknown part $\delta t_{d}$ that is a possible positive or negative value. To save computation, constrained Kalman filtering is not necessary. Instead, interpolation and quaternion Slerp explained in Section 3.2.1 are tractable.

From the definition of time delays presented in Section 3.1, the total time delay is not estimated as negative. For example, if the estimated delay is negative (i.e., an exceeded index), estimation is impossible since this case is forecasting states or obtaining measurements from the future, so the total delay has to be bounded by zero. Moreover, in the sequential measurement update, if estimated time delay $\delta t_{d}$ is larger than the sampling time of the IMU, $\Delta t_{\mathrm{IMU}}$, then we indicate another slot in the delay buffer. Algorithm 1 is a summarized algorithm of the overall processes of the latency compensated VIO.
**Algorithm 1** The Latency Compensated VIO**Require:**${\hat{x}}_{0}^{+}\left(={\hat{x}}_{V_{0}}^{+}\right),P_{0}^{+},Q,R,\;P^{\mathrm{dly}}\left(=P_{0}^{+}\right),\Phi^{\mathrm{crs}}\left(=I\right),\;\chi^{2}$1:**for**k=1:T**do**2:   **if** new IMU packet arrival **then**
3:*Time Update*: 4:    ${\dot{\hat{x}}}_{V}=f\left(\;{\hat{x}}_{V_{k-1}}^{+},\;a_{\mathrm{raw}},\omega_{\mathrm{raw}}\;\right)$                             ▹ static features5:    Numerically integrate with $\Delta \;t_{\mathrm{IMU}}\left(=t_{k}-t_{k-1}\right)$6:    ${\hat{x}}_{k}^{-}={\hat{x}}_{k-1}^{+}+\int_{t_{k-1}}^{t_{k}}\dot{\hat{x}}\left(\tau \right)\;d\tau$7:    $\Phi_{k-1}=\exp\left(\int_{t_{k-1}}^{t_{k}}A\left(\tau \right)\;d\tau \right)$                               ▹$A=\frac{\partial f}{\partial x}{|}_{{\hat{x}}_{V}}$8:    ${\left(P_{vv}\right)}_{k}^{-}=\Phi_{k-1}\;{\left(P_{vv}\right)}_{k-1}^{+}\;\Phi_{k-1}^{T}+B_{k-1}\;Q_{v}\;B_{k-1}^{T}$                         ▹$B=\frac{\partial f}{\partial \eta }{|}_{{\hat{x}}_{V}}$9:    $P_{k}^{-}=\left[\begin{array}{cc} {\left(P_{vv}\right)}_{k}^{-} & \Phi_{k-1}\;{\left(P_{vf}\right)}_{k-1}^{+}\\ {\left(P_{fv}\right)}_{k-1}^{+}\;\Phi_{k-1}^{T} & {\left(P_{ff}\right)}_{k-1}^{+}+Q_{f} \end{array}\right]$10:    Store the state estimates into the delay buffer11:    **if** during delay period **then**12:      $\Phi^{\mathrm{crs}}\leftarrow \Phi_{k-1}\;\Phi^{\mathrm{crs}}$                                  ▹ recursive13:    **else**14:      $P^{\mathrm{dly}}\leftarrow P_{k}^{-}$15:      $\Phi^{\mathrm{crs}}\leftarrow I$16:    **end if**17:   **end if**18:   **if** new vision data packet arrival **then**
19:    Compute index ${\hat{d}}^{-}$ of delay20:    Interpolate using the state estimates from the buffer21:    **for**
j=1:♯ of observed features **do**22:      **if** new feature **then**23:*Feature Initialization*:24:        ${\hat{p}}_{f_{j}/i}=g_{j}\left(\;{\hat{x}}_{k-\hat{d}},\;y_{j}\;\right)$                     ▹ least-squares minimization25:        Augment state if feature is valid                         ▹ if positive depth26:        ${\hat{x}}_{k}\overset{\mathrm{aug}}{\Rightarrow }{\left[\begin{array}{cc} {\hat{x}}_{k}^{T} & {\hat{p}}_{f_{j}/i}^{T} \end{array}\right]}^{T}$27:        $P^{\mathrm{dly}}\overset{\mathrm{aug}}{\Rightarrow }\left[\begin{array}{cc} P^{\mathrm{dly}} & P^{\mathrm{dly}}\;J_{j}^{T}\\ J_{j}\;P^{\mathrm{dly}} & J_{j}\;P^{\mathrm{dly}}\;J_{j}^{T}+P_{f_{j_{\mathrm{new}}}} \end{array}\right]$                  ▹$J_{j}=\frac{\partial p_{f_{j}/i}}{\partial x}{|}_{{\hat{x}}_{k-\hat{d}}}$28:        $P_{k}\overset{\mathrm{aug}}{\Rightarrow }\left[\begin{array}{cc} P_{k} & 0\\ 0 & \hat{d}\;Q_{f} \end{array}\right]$
29:        Prune state vector if exceed maximum30:      **else**                                     ▹ tracked feature31:*Measurement Update*: 32:        Update if gating test is passed                         ▹$r_{j}^{T}S_{j}^{-1}r_{j}\overset{?}{<}\chi_{j}^{2}$33:        $r_{j}=y_{j}-h_{j}\left({\hat{x}}_{k-\hat{d}}\right)$34:        $S_{j}={\left(C_{j}\right)}_{k-\hat{d}}\;P^{\mathrm{dly}}\;{\left(C_{j}\right)}_{k-\hat{d}}^{T}+R$                    ▹${\left(C_{j}\right)}_{k-\hat{d}}=\frac{\partial h_{j}}{\partial x}{|}_{{\hat{x}}_{k-\hat{d}}}$
35:        $P^{\mathrm{crs}}\leftarrow \left[\begin{array}{cc} \Phi^{\mathrm{crs}}\;P_{vv}^{\mathrm{dly}} & \Phi^{\mathrm{crs}}\;P_{vf}^{\mathrm{dly}}\\ P_{fv}^{\mathrm{dly}} & P_{ff}^{\mathrm{dly}} \end{array}\right]$36:        $K_{j}^{\mathrm{crs}}=P^{\mathrm{crs}}\;{\left(C_{j}\right)}_{k-\hat{d}}^{T}\;S_{j}^{-1}$37:        $\Delta \;{\hat{x}}_{k}=+K_{j}^{\mathrm{crs}}\;r_{j}$                             ▹$\Delta {\hat{t}}_{d}\overset{?}{<}\Delta t_{\mathrm{IMU}}$38:        $\Delta \;P_{k}=-K_{j}^{\mathrm{crs}}\;{\left(C_{j}\right)}_{k-\hat{d}}\;{\left(P^{\mathrm{crs}}\right)}^{T}$39:        Sequentially update the buffer40:        $K_{j}^{\mathrm{dly}}=P^{\mathrm{dly}}\;{\left(C_{j}\right)}_{k-\hat{d}}^{T}\;S_{j}^{-1}$41:        $\Delta \;{\hat{x}}_{k-\hat{d}}=+K_{j}^{\mathrm{dly}}\;r_{j}$
42:        $\Delta \;P^{\mathrm{dly}}=-K^{\mathrm{dly}}\;{\left(C_{j}\right)}_{k-\hat{d}}\;P^{\mathrm{dly}}$43:      **end if**44:    **end for**45:    Store index ${\hat{d}}^{+}$ of the posterior estimated delay 46:    $P^{\mathrm{dly}}\leftarrow P_{k}^{+},\;\Phi^{crs}\leftarrow I$
47:    Erase used slots in the delay buffer48:   **end if**49:**end for**

## 5. Results

This section provides results from Monte Carlo trials and flight datasets testing. First, in a simulation environment, since we can set true time-delay values at measurements, we test if the proposed framework estimates the actual time delays accurately. The next subsection presents the performance of the proposed approach by testing with benchmark flight datasets for validating whether it solves real-world problems.

### 5.1. Monte Carlo Simulations of a Simple Example

To show actual-time delays being estimated accurately, this section simulates a simple example problem by 100 Monte Carlo trials. The vehicle and measurement models of this simulation are direct from Lee and Johnson’s previous work [43]. The models are a second-order dynamic system with a non-delayed speed measurement and two delayed bearing angles measured from each location of two stations. From Equation (Equation 49), variance $Q_{d}$ value of this simulation is $0.25\;s^{2}$.The actual time delay of the delayed measurements in this simulation is 0.9s, and this value is identical to 18 delayed samples since the propagation rate of the simulation is 0.05s. Monte Carlo simulations estimate the values of time delays, shown in Figure 7.

Figure 7a shows that the estimated delay rapidly converges to the true delay value. That is, the estimation error of the delayed samples gradually decreases toward zero. Moreover, we may wonder whether the latency compensated filtering algorithm works when the delay is not static. See Figure 7b, which illustrates a response to a change in time delay. Although the values of unknown delays vary over time, estimation resulting from the online calibration method converges to true delay values.

### 5.2. Flight Datasets’ Test Results

To validate the reliability of the proposed approach for estimating states and unknown delay values, we test one of the benchmark datasets, so-called “EuRoC MAV datasets [63].” The visual-inertial sequences of the datasets were recorded onboard a micro aerial vehicle while a pilot manually flew around indoor Vicon environments. For more details, see reference [63]. Although the datasets include noise model parameters from the IMU at rest, we need to tune each variance of process noise covariance *Q* for the best performance. Likewise, to estimate the unknown part of time delays, we set the standard deviation of random walk $\eta_{d}$ in Equation (Equation 48) as $1.0\times 10^{-5}$ since the order of this value is set to the same order of the smallest value among the provided noise parameters.

Given that datasets provide various levels of challenging sequences such as faster motion and poor illumination in each environment, to articulate the significance of time delays defined in Section 3.1, we select two datasets of slow motion, called “EuRoC V1 Easy,” and fast motion, called “EuRoC V1 Medium.” Since the vehicle in the medium dataset maneuvers twice as fast, we hypothesize that the time delays have greater impact on the navigation solution of the medium dataset. Algorithms of image processing and filtering are developed under the robot operating system (ROS) [64], given that IMU data and images from the stereo camera are also subscribed under the ROS, shown in Figure 8.

The simplest solution to the estimation problem of the given datasets is to run the baseline in Section 3.2.1 that corrects only Jacobians and residual. However, the novel latency compensated filter described in Algorithm 1 compensates for delayed measurements at the time when the vision data are fused at the filter and estimate the refined state and the delay values. This compensated filtering follows the processes of all three corrections, shown in Figure 6.

The EKF estimates relative location from a starting point. Since we do not know the exact absolute location of origin of given datasets, to compare with ground truth data given in the datasets, certain evaluation error metrics such as so-called “absolute trajectory error [65]” are required.

After applying the absolute trajectory error, Figure 9 illustrates the top-down view of the estimated flight trajectory of the medium dataset. Figure 10 exhibit estimated *x*, *y*, *z* position and their estimation errors.

All estimation errors are bounded within each standard deviation σ bounds. We should expect significant time correlation in error plots and a generally growing error covariance for vision-aided inertial navigation problems like this one. Conceptually, position error gets “locked in” and to the extent new features are being mapped the position error will tend to grow with the length of the trajectory. Starting from the noise model parameters reported for the datasets, the compensated filter is a well-tuned estimator—the performance of doing runs with ×3 or ×10 (/3 or /10) multiplier on the *R* term used in the filter is worse for all of those, shown in Table 2. In other words, the fact that using those multipliers shows larger root mean square (RMS) estimation errors indicates that our approach is a well-tuned filter.

Figure 11 shows the advantages of each correction in the latency compensated VIO by comparing it with the baseline and the covariance correction. The baseline discards cross-covariance and unknown part of the delays, and, although the latency compensated filter might increase the computational effort of the entire system, it significantly improves the accuracy of estimation.

Unlike either the baseline or the covariance correction, the latency compensated filter calibrates the unknown part of time delays. Figure 12 shows that estimation resulting from the compensated filter converges to a certain, final delay value, and its variance rapidly decreases although initial uncertainty is high.

As shown in Figure 13, the average of estimated total delays is around 45ms that could generate about 4cm drift and offset during the delay period when the vehicle fly at 0.91m/s average speed.

When readable delay values are negative, the timestamps of images might indicate wrong pairs or packets.

Table 3 lists the RMS position errors of cases for sensitivity analysis. Approximately known part of time delays introduced in Section 3.2 are either fixed ${\bar{t}}_{d}$ by tuning or readouts ${\bar{t}}_{d_{\mathrm{raw}}}$, which is the difference of readable timestamps of current IMU and image. In addition, we can directly estimate entire parts of time delays without information on the approximately known part. For another case, using the final value of the estimated unknown part of time delays, we add a fixed ${\bar{\delta t}}_{d}$ to the total delays at every time. However, this case might not work when the delay is varying, and we can know the final value only after running the proposed filter. In other words, before applying the compensation filtering, fixed ${\bar{\delta t}}_{d}$ is still unknown. The estimation results from the latency compensated VIO approach depict the influence of the delays and the effectiveness of the corrections in the sensor fusion of the lagged measurements.

Fast motion datasets are more sensitive to time delays since the improvement is larger when applied to those datasets.

Although numerous researchers have explored the VIO of the EuRoC datasets, few of them thoroughly considered measurements with unknown time delays. Table 4 reveals that the proposed estimator, the latency compensated VIO, outperforms the existing state-of-the-art methods, called “S-MSCKF” and “SVO+MSF” in which stereo is available.

## 6. Discussion

This study develops a practical extended Kalman filter (EKF)-based visual-inertial odometry (VIO) accounting for vehicle-feature correlations; that is, we develop tightly-coupled VIO for autonomous flight of unmanned aerial vehicles (UAVs). In particular, this paper has presented the development of a reliable and accurate filtering scheme for measurements with unknown time delays. We define time delays of vision data measurements in VIO. For compensating delayed measurements and estimating unknown delay values, this paper presents latency compensated filtering that includes state augmentation, interpolation, and residual, Jacobian, covariance corrections. The optimality of the three corrections and the observability of the state augmentation are validated; in other words, the appendix shows that the proposed latency compensated filter results in optimal estimates as if there were no delays in the data.

We test the performance of VIO employing the latency compensated filtering algorithms in the benchmark flight datasets for comparison to other state-of-the-art VIO algorithms. Results from flight datasets testing show that the novel navigation approach in this paper improves the accuracy and reliability of state estimation with unknown time delays in VIO. With the latency compensated filtering, the root mean square (RMS) errors of estimation are decreased. In particular, we show improved accuracy of our method over previous approaches for state estimation in the fast motion datasets.

The overall approach in this document can be easily employed in other filter-based, sensor-aided inertial navigation frameworks and is suitable to monocular VIO although this study uses a stereo camera to showcase the methods. Although the reliability and robustness of this study are validated by testing benchmark flight datasets, validating with other datasets is of interest.

## Figures and Tables

**Figure 1 sensors-20-02209-f001:**
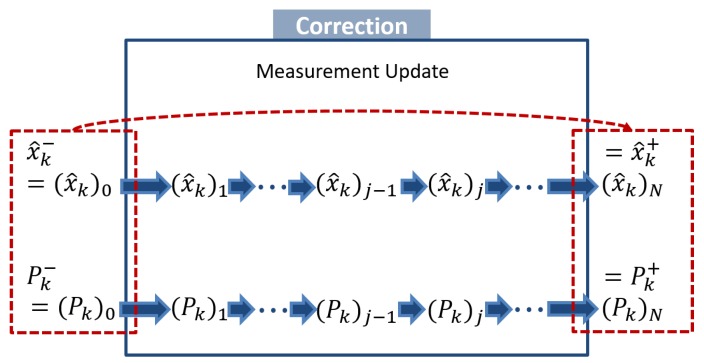
A schematic of the sequential measurement update.

**Figure 2 sensors-20-02209-f002:**
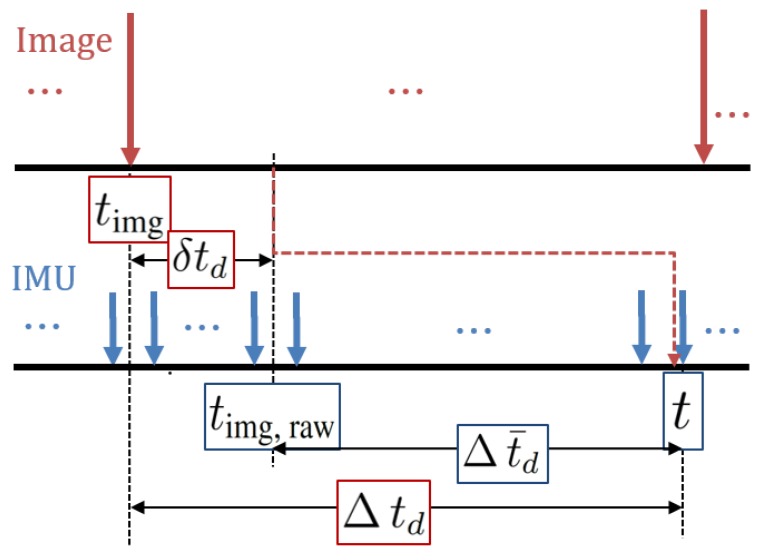
Data streams of the IMU and the delayed vision data.

**Figure 3 sensors-20-02209-f003:**
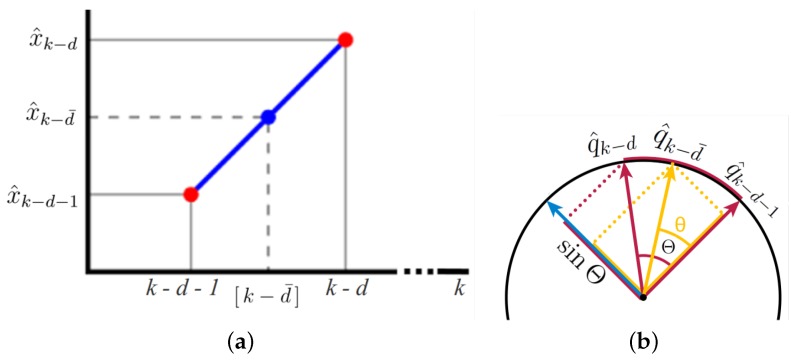
Examples of interpolation and slerp. (**a**) Linear interpolation, (**b**) Quaternion slerp.

**Figure 4 sensors-20-02209-f004:**
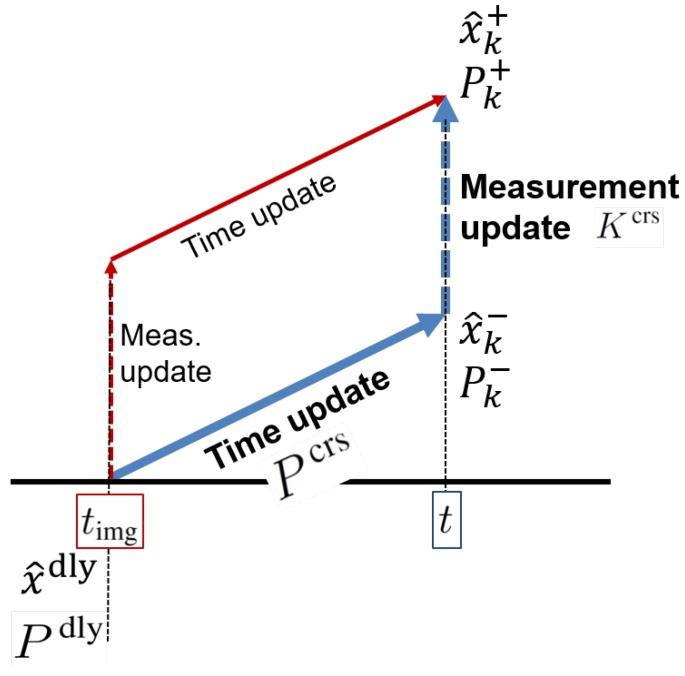
A schematic of a modified measurement update using covariance correction.

**Figure 5 sensors-20-02209-f005:**
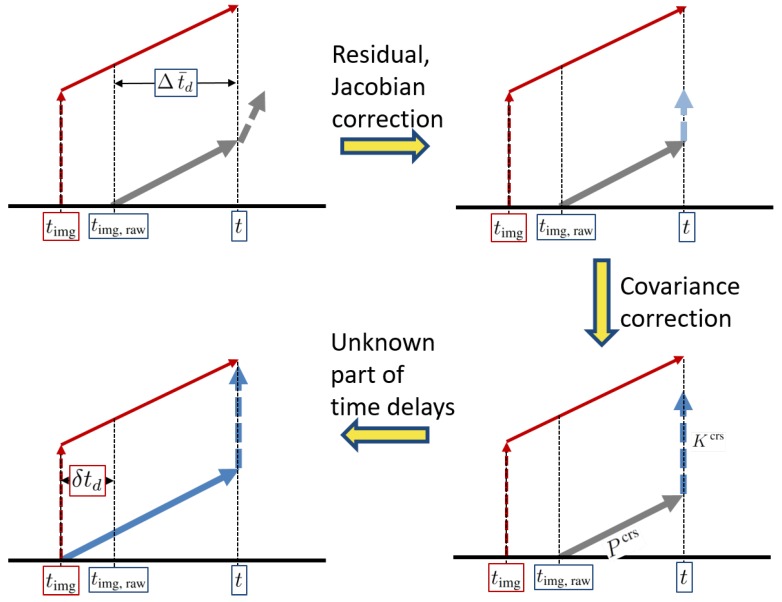
Three corrections in the latency compensated VIO.

**Figure 6 sensors-20-02209-f006:**
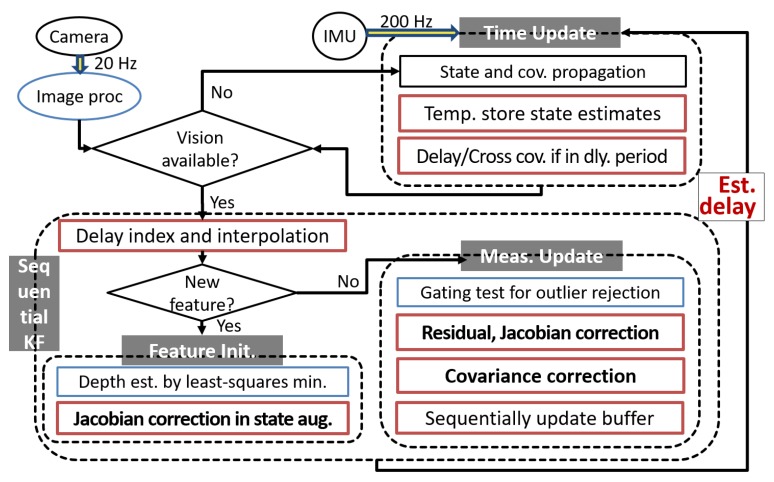
A flow chart of the overall process of the latency compensated VIO.

**Figure 7 sensors-20-02209-f007:**
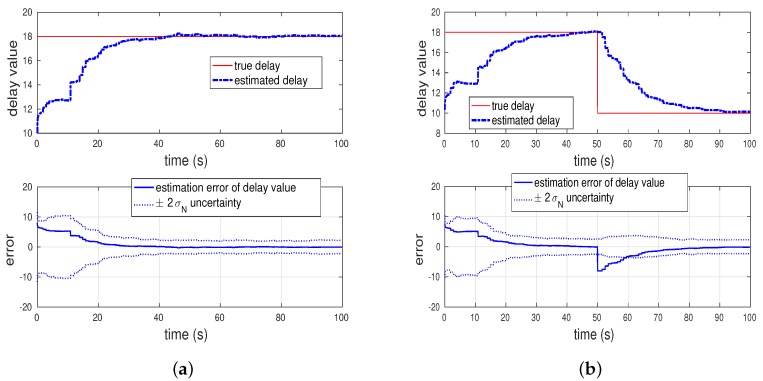
Estimation of total delays in simulation. (**a**) A static delay, (**b**) Varying delays.

**Figure 8 sensors-20-02209-f008:**
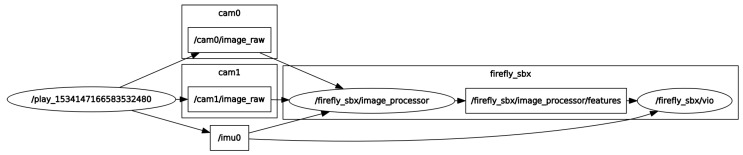
ROS rqt graph of the latency compensated VIO linked to the EuRoC dataset.

**Figure 9 sensors-20-02209-f009:**
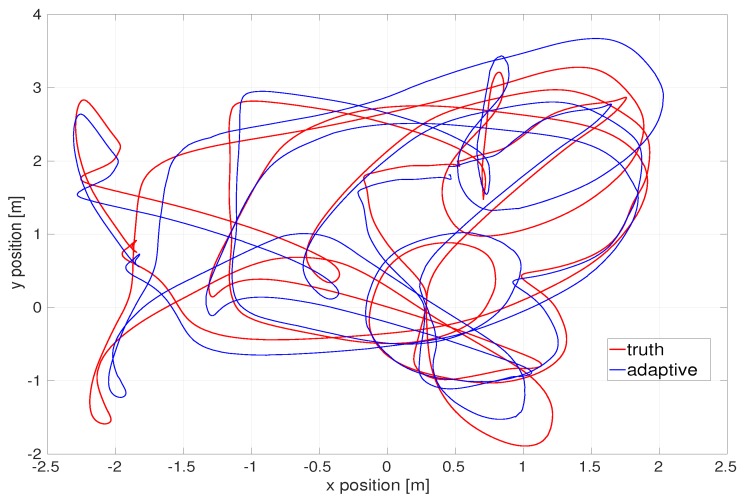
Top-down view of flight trajectory of the EuRoC V1 medium dataset by the latency compensated VIO.

**Figure 10 sensors-20-02209-f010:**
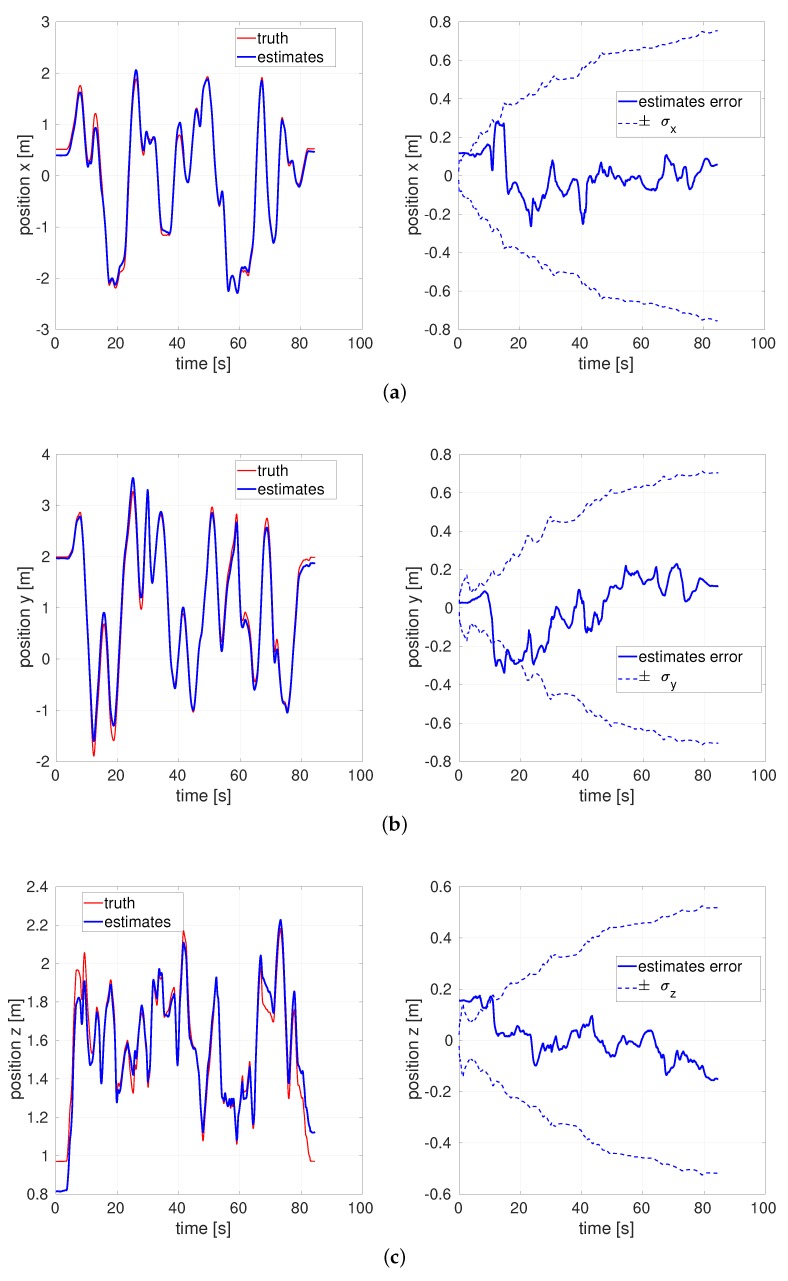
Position and estimation error of the EuRoC V1 medium dataset by the latency compensated VIO. (**a**) Position *x*, (**b**) Position *y*, (**c**) Position *z*.

**Figure 11 sensors-20-02209-f011:**
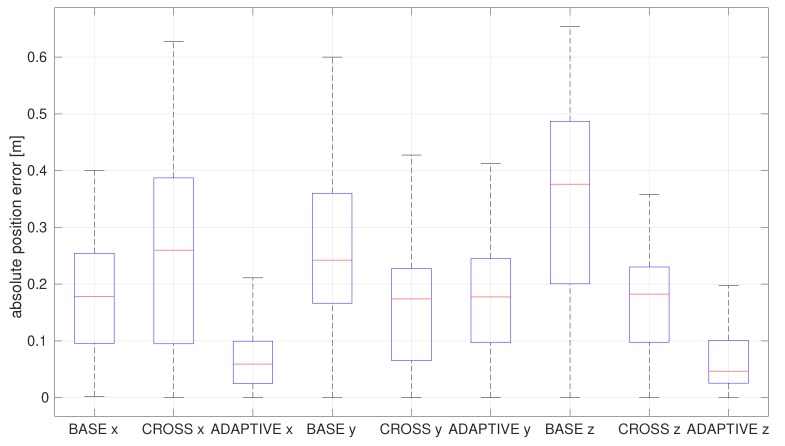
Box plot of absolute estimation error of position of the EuRoC V1 medium dataset by the latency compensated VIO.

**Figure 12 sensors-20-02209-f012:**
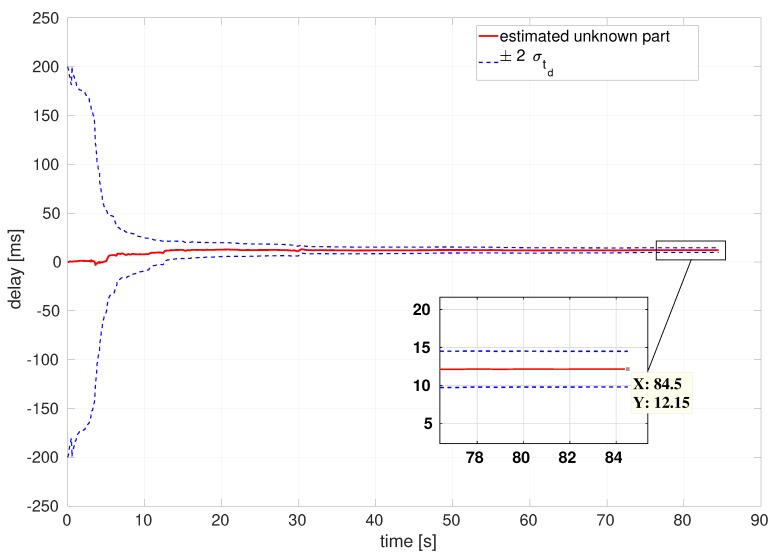
Estimation of unknown part of time delays of the EuRoC V1 medium dataset.

**Figure 13 sensors-20-02209-f013:**
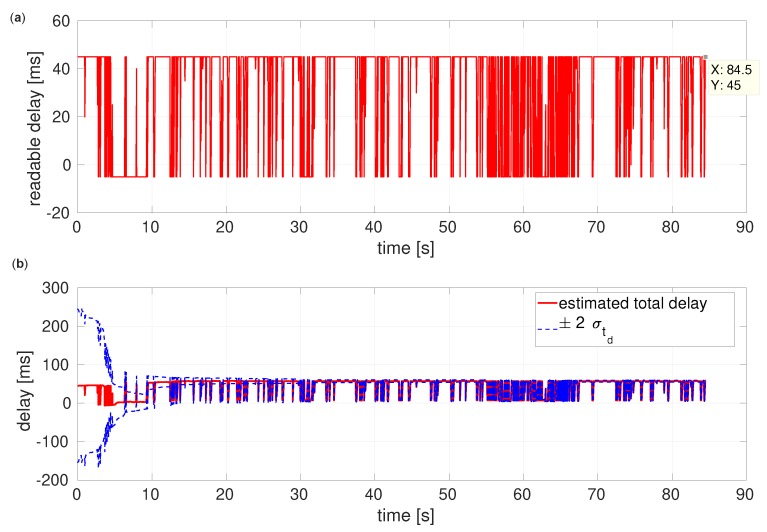
Estimation of time delays of the EuRoC V1 medium dataset. (**a**) Readable delays, (**b**) Estimated total delays.

**Table 1 sensors-20-02209-t001:** State-of-the-art Visual-Inertial Odometry.

Name		ROVIO [15]	VINS-MONO [16]	SVO +MSF [17]	Alternating Stereo VINS [18]	S-MSCKF [19]	OKVIS [20]
Monocular		×	×	×			
Stereo					×	×	×
Indirect			×		×	×	×
Semi-direct				×			
Direct		×					
Loosely-Coupled				×			
Tightly-Coupled		×	×		×	×	×
Optimization-based			×				×
Filtering-based		×		×	×	×	
Open-source		×	×	×		×	×

**Table 2 sensors-20-02209-t002:** Indication that the latency compensated VIO is well-tuned for EuRoC V1 medium dataset.

Multiplier on *R*		/10	/3	1	×3	×10
RMS error [m]		1.5096	0.1969	0.1619	0.2636	0.2850

**Table 3 sensors-20-02209-t003:** Sensitivity analysis in RMS position error [m] of latency compensated VIO.

	Dataset		EuRoC V1 Easy *Slow Motion* 0.41 m/s, 16.0 deg/s		EuRoC V1 Medium *Fast Motion* 0.91 m/s, 32.1 deg/s	
Method			Cross-Cov OFF	Cross-Cov ON	Cross-Cov OFF	Cross-Cov ON
Fixed ${\bar{t}}_{d_{\mathrm{const}}}$			0.3376	0.2677	0.4644	0.3135
Entirely Estimated ${\hat{t}}_{d}$			0.2282	0.2406	0.4734	0.3538
Readouts ${\bar{t}}_{d}$	+ N/A		0.2558	0.2032	0.4163	0.3121
	+ Fixed ${\bar{\delta t}}_{d}$		0.2869	0.2285	0.3281	0.2218
	+ Estimated ${\hat{\delta t}}_{d}$		0.2019	**0.1461**	0.3353	**0.1619**

**Table 4 sensors-20-02209-t004:** Comparison with other methods in RMS position error [m] of latency compensated VIO

	Dataset	EuRoC V1 Easy	EuRoC V1 Medium
Method		*Slow Motion* 0.41 m/s, 0.28 rad/s	*Fast Motion* 0.91 m/s, 0.56 rad/s
Latency Compensated VIO		0.1461	0.1619
S-MSCKF (stereo-filter)		0.34	0.20
SVO+MSF (loosely-coupled)		0.40	0.63

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
