# Peer review of "Latency Compensated Visual-Inertial Odometry for Agile Autonomous Flight"

_sensors, 2020, doi:10.3390/s20082209_

Round 1

Reviewer 1 Report

A latency compensated filtering method was proposed for Visual Inertial Odometry in this manuscript. Comments for revising:

  1. Although some examples were given to show the existence of uncertainty delay, but the factors that would cause delay and its variance can be summarized and categorized, for example, communication delay would be one kind of delay sources? and which delay sources will be the key delay or delay uncertainty contributors?
  2. It seems that there is a boundary for some delay sources, how can the boundaries be used in the estimation process?
  3. The application background was not described clearly in the abstract and introduction part, does the proposed method aim at multi sensor scenarios?
  4. I think the explanations for tightly-coupled VIO and loosely-coupled VIO, Optimization-based and Filtering-based VIO are needed for better understanding.
  5. I could not find “y” in figure 1, it's x ?
  6. How to get 3D position from camera frames?
  7. What simulation platform was used? and how to get the true delay?
  8. How to understand the convergence performenc of the estimation for x,y,z position errors in Figures10?

Reviewer 2 Report

The paper addresses the problem of estimation of a position of an agile autonomous aerial vehicle. Specifically, a novel algorithm for latency compensation of visual-inertial odometry based on an extended Kalman filter is introduced. The key idea lies in direct estimation of the time delay as an additional state of the system.

Generally, the paper is well written and I like it. The motivation is clear and gives the reader a good reason why to read the paper. The state of the art is, according to my knowledge, complete and contains all relevant works, while the contribution is nicely stated with the state of the art. The method itself is described in a sufficient level of detail. The presented experimental results are convincing as the performance of the method is demonstrated on the standard datasets and the comparison with the state of the art techniques is done. The results show that the introduced method outperforms the state of the art methods. I found only two typos:

- line 364: measurements at thE time

- line 366: processes of all three correctionS

Overall, the paper can be accepted as it is and I believe that it will be a solid contribution to the journal.

Reviewer 3 Report

This paper proposes three correction techniques into state estimation to combine the camera input and extended Kalman filter-based Visual Inertial Odometry systems. These three steps are clearly explained in the manuscript and the result mirrors the claim that authors make.

Please modify the abstract to explicitly mention what is new about this study and comment on results. I like Section 1.2 and the essence of this section should be mentioned in the abstract.

The results section of the paper shows the output from Monte Carlo trials and EuRoC MAV. Please explain/outline the strategy or rationale behind choosing these under section 5 before 5.1.
